# Genome-Wide Association Study in Mexican Holstein Cattle Reveals Novel Quantitative Trait Loci Regions and Confirms Mapped Loci for Resistance to Bovine Tuberculosis

**DOI:** 10.3390/ani9090636

**Published:** 2019-08-30

**Authors:** Sara González-Ruiz, Maria G. Strillacci, Marina Durán-Aguilar, Germinal J. Cantó-Alarcón, Sara E. Herrera-Rodríguez, Alessandro Bagnato, Luis F. Guzmán, Feliciano Milián-Suazo, Sergio I. Román-Ponce

**Affiliations:** 1Doctorado en Ciencias Biológicas, Universidad Autónoma de Querétaro, Avenida de las Ciencias S/N Juriquilla, Delegación Santa Rosa Jáuregui, Querétaro C.P. 76230, Mexico; 2Department of Veterinary Medicine, Università degli Studi di Milano, Via Trentacoste, 2, 20134 Milano, Italy; 3Facultad de Ciencias Naturales, Universidad Autónoma de Querétaro, Avenida de las Ciencias S/N Juriquilla, Delegación Santa Rosa Jáuregui, Querétaro C.P. 76230, Mexico; 4Centro de Investigación y Asistencia en Tecnología y Diseño del Estado de Jalisco A.C., Guadalajara C.P. 44270, Mexico; 5Centro Nacional de Recursos Genéticos, INIFAP, Tepatitlán de Morelos 47600, Mexico; 6Centro Nacional de Investigación Disciplinaria en Fisiología y Mejoramiento animal, INIFAP, SAGARPA, Km. 1 Carretera a Colón, Ajuchitlán, Colón, Querétaro C.P. 76280, Mexico

**Keywords:** bovine tuberculosis resistance, DNA pooling, SNP, QTL, genome-wide association study

## Abstract

**Simple Summary:**

Bovine tuberculosis is an infectious disease of cattle caused by *Mycobacterium bovis* characterized by the formation of tubercles in any organ or tissue. Bovine tuberculosis represents a significant veterinary and public health problem in many parts of the world. It is zoonotic, transmitted to humans through consumption of infected milk and other cattle products. Although many factors influence infection and progression of the disease, there must be an important host genetic component that explains why some animals get sick and others remain healty. We present evidence of genetic variants associated with resistance to tuberculosis in Mexican Holstein dairy cattle using a case-control approach with a selective DNA pooling. Here, we identified novel quantitative trait loci regions harboring genes involved in *Mycobacterium* spp. immune response. This is a first screening about resistance to tuberculosis infection on Mexican Holstein cattle based on a dense single nucleotide polymorphism chip. The identified genes belong to both, the already known, and the undisclosed quantitative trait loci regions.

**Abstract:**

Bovine tuberculosis (bTB) is a disease of cattle that represents a risk to public health and causes severe economic losses to the livestock industry. Recently, genetic studies, like genome-wide association studies (GWAS) have greatly improved the investigation of complex diseases identifying thousands of disease-associated genomic variants. Here, we present evidence of genetic variants associated with resistance to TB in Mexican dairy cattle using a case-control approach with a selective DNA pooling experimental design. A total of 154 QTLRs (quantitative trait loci regions) at 10% PFP (proportion of false positives), 42 at 5% PFP and 5 at 1% PFP have been identified, which harbored 172 annotated genes. On BTA13, five new QTLRs were identified in the *MACROD2* and *KIF16B* genes, supporting their involvement in resistance to bTB. Six QTLRs harbor seven annotated genes that have been previously reported as involved in immune response against *Mycobacterium* spp: BTA (*Bos taurus* autosome) 1 (*CD80*), BTA3 (*CTSS*), BTA 3 (*FCGR1A*), BTA 23 (*HFE*), BTA 25 (*IL21R*), and BTA 29 (*ANO9* and *SIGIRR*). We identified novel QTLRs harboring genes involved in *Mycobacterium* spp. immune response. This is a first screening for resistance to TB infection on Mexican dairy cattle based on a dense SNP (Single Nucleotide Polymorphism) chip.

## 1. Introduction

Bovine tuberculosis (bTB), caused by *Mycobacterium bovis* is a chronic infectious disease characterized by granulomas in affected tissues [1,2]. *M. bovis* infects a wide range of mammalian hosts, domestic and wildlife species, and humans; therefore, it is a risk to public health [3]. It has been estimated that nearly 10 million people are affected by tuberculosis worldwide every year, and that the proportion of cases due to *M. bovis* in humans during the last two decades was from 0.5% to 13%, depending on the study population [4,5,6]. Additionally, bTB causes economic losses to the livestock industry: infected animals have poor production performance, die or are disposed of prematurely [7,8]. Cattle TB is considered the fourth most significant livestock disease in terms of impact on human health in developing countries, including risks to species other than cattle and the wildlife species [9]. The disease persists in livestock in spite of the on-going eradication program that has been established. The program relies on a test-and-slaughter strategy in herds of cattle, and carcass inspection at abattoirs [10].

Recently, genetic studies like genome-wide association studies (GWASs) have greatly improved the understanding of complex diseases identifying thousands of disease-associated genomic variants [11]. Evidence suggests that genetic variation and resistance to bTB exists in many species, including humans, mice, deer and cattle [12,13]. Heritability values estimated on UK and Irish cattle populations have shown that individual variability for host resistance to TB has a genetic basis [14,15]. Other studies have also shown genetic variation for resistance of cattle to TB [15]; higher resistance has been reported in *Bos taurus indicus* compared to *Bos taurus* [16,17].

In Mexico, bTB is still an endemic disease, and the availability of genomic tools, such as high-density SNP (Single Nucleotide Polymorphism), allow disclosing QTL (quantitative trait loci) regions harboring genes involved in the immune response against TB, as previously reported in different cattle populations. Several studies have in fact identified genetic loci associated with bTB resistance. They included polymorphisms in candidate genes like *SLC11A1* in African Zebu cattle [18], *TLR1* in Chinese Holsteins [19], SNP on BTA23 in Irish dairy herds [20], and three other genetic loci on BTA2 and 13 were also associated [21]. A GWAS involving Irish Holsteins identified a genomic region in BTA22 containing the taurine transporter gene *SLC6A6*, which was suggestively associated with resistance [22]. In a case-control study, GWAS used in *Mycobacterium avium* subsp. paratuberculosis (MAP) identified chromosomal regions (BTA9, BTA11 and BTA12) associated with this disease; and provides evidence of genetic loci involvement in humoral response to MAP [23].

Therefore, the aim of this study was to identify QTL regions involved in resistance to TB in Mexican dairy cattle using a GWAS case-control approach with a selective DNA pooling experimental design.

## 2. Materials and Methods

This project was approved by the Bioethics Committee of the Natural Sciences Department of the Autonomous University of Queretaro under registry number 29FCN2016.

### 2.1. Tissue Samples

Tissue samples were collected from carcasses at slaughterhouses in the States of Jalisco and Aguascalientes. These two states are located in central Mexico where the within-herd prevalence of tuberculosis in dairy cattle is about 16% [24,25]. Animals slaughtered were Holstein cows from small family-run herds with an average size of 70 head.

Even when all lymph nodes and internal organs were checked for the presence of lesions, tissue samples selected were taken only from lymph nodes in head (retropharyngeal), thorax (tracheobronchial and mediastinal), abdomen (mesenteric), and lungs. Tissue samples were collected both from animals with visible lesions and from animals with no visible lesions at carcass inspection. After collection, tissue samples were immediately placed in a cooler with ice and taken to the laboratory where they were kept at −20 °C until analysis. Hair samples were taken from the ear in the same animals as a source of DNA for SNP genotyping. Epidemiological data for each animal included herd and States of sampling, sex, age, and the organ affected, and was used to evaluate homogeneity of prevalence of bTB across different geographical areas of sample collection.

### 2.2. Bacteriological Analysis

All tissue samples were cultured in Stonebrink and Lowenstein-Jensen media with pyruvate for the isolation of *M. bovis* (Figure 1). Briefly, tissue samples were first surface-sterilized with 1:1000 solution of sodium hypochlorite, and then macerated and decontaminated with a 10% solution of hydrochloric acid as previously reported [25,26].

### 2.3. Experimental Design, DNA Extraction, Pooling and Genotyping

A total of 375 biological samples were included in the study, 150 cases (tissue samples with visible lesions and culture positive) and 225 controls (tissue samples with no visible lesions and culture negative), collected from carcasses. All samples were from the same geographic area and, in some cases, from the same herd, to ensure similar level of exposure to the pathogen as previously described [24,25].

A selective DNA pooling design [27] was used in this study to identify QTL regions associated with resistance to bTB in a case control study. This experimental design has been shown to be effective, appropriate, powerful to perform association studies, and highly accurate compared with experimental designs using individual-sample genotyping [28,29]. Selective DNA pooling has been extensively used in GWAS mapping studies in livestock for mapping QTL in quantitative traits [29,30], and in case control studies in cattle [31,32].

With the advent of dense SNP chip arrays, the source of variation related to the experimental design and the methodology to control and accurately account for it has been discussed and established [33,34].

DNA extraction was performed using a commercial kit (Wizard^®^ Genomic DNA Purification Kit, Thermo Fisher Scientific, Waltham, MA, USA) following the manufacturer recommendations. Quantity and quality control on each DNA sample was performed by spectrophotometry with NanoDrop™ 2000 equipment (Thermo Fisher Scientific). Integrity of the DNA was determined by electrophoresis on a 1% agarose gel pre-stained with GelRed^®^ Nucleic Acid Gel Stain (Biotium, Fremont, CA, USA).

All DNA samples were normalized at a concentration of 50 ng/µL. DNA pools were then built by taking equivalent amounts of volume from each DNA sample, thus the final concentration for each pool was 50 ng/µL according to Illumina array requirements.

A total of 75 DNA samples were used in building each pool, and biological, technical and array replicates were designed as suggested [33,34]. Pools for cases were composed of two independent groups of 75 animals (average age of 47 ± 1.8 months and the proportion was 75% females and 25% males) positive for visible lesions at carcass inspection and positive for *M. bovis* isolation by culture (Figure 1). These two pools represent two independent biological replicates. The pools for controls were composed of three independent groups of 75 animals (average age of 43 ± 2.8 months and the proportion was 62% females and 38% males) with no visible lesions and negative for *M. bovis* isolation by culture. Each of the pools was produced in two replicates to account for possible errors in pooling the individuals (technical pooling replicates, i.e., Pool_Rep “_A” and “_B” in Table 1) and respectively genotyped three times to account for array technical error (technical array replicates, i.e., Array_Rep “_1”, “_2”, and “_3” in Table 1).

The 10 pools (biological and technical replicates) were processed in three array replicates each, on the Illumina BovineHD BeadChips (777,962 SNP), following the Infinium protocol obtaining a total of 30 sets of B-allele frequency for each SNP. SNPs position was determined according to the UMD 3.1 bovine assembly.

### 2.4. Statistical Analysis of Pool

The B-allele frequencies (BAF) values for each SNP were obtained from the self-normalization algorithm of Illumina BeadStudio software^®^ for each of the three arrays technical replicates of the 10 pools. The BAF is a very accurate measure of the frequency of the alleles in all individuals together in a pool as previously reported [29].

First, a quality control was performed at array technical replicate comparing the standard deviation (SD) distribution of B-allele frequencies among each triplet of array technical replicates. Two array-replicates, one case and one control, were eliminated from the analysis because the value of their B-allele frequency diverged from the other two technical array-replicates. Second, a quality control on the BAF estimation was performed at SNP level as follows: the SD among BAF from the replicate assays (biological, pool and array technical replicates) within cases and controls was calculated, and the markers showing the largest 10% SD were excluded from the analysis. Finally, only SNPs with minor allele frequency (MAF) ≥0.05 were retained. After editing, a total of 438,555 SNPs (of which 10,034 were on BTX (*Bos taurus* X autosome) were used in the association analysis.

GWAS was performed comparing at each marker the allele frequencies obtained for the cases pools with those obtained in the control pools for each marker (averaged over replicates within case and within control) according to the selective DNA pooling (SDP) design and methods as described in detail [31,32]. GWAS was performed after excluding monomorphic SNPs, SNPs mapped on BTY, mitochondrial SNPs, and SNPs without chromosome position.

A single-marker test for marker-trait association was used, and the *p*-value for each marker calculated as: Ztest = Dtest/SD (Dnull)(1) where: Dtest is the difference of the B-allele frequencies means among tails, and Dnull is the difference of the B-allele frequencies means within tails.

### 2.5. Quantitative Trait Loci Region Definition

The nominal P values at different PFP (proportion of false positives) thresholds, i.e., 1%, 5% and 10%, have been identified according to Fernando et al. [34,35], and the corresponding −log10 (*p*-value) calculated, resulting: (i) for PFP at 1%, 4.58; (ii) for PFP at 5%, 3.17; (iii) for PFP at 10%, 2.53. As in Lipkin et al. [31], moving averages of −log10 (*p*-values) were calculated considering a window of 16 SNP markers, corresponding to an average-window-size of about 100 Kb. As shown in Lipkin et al. [31], PFP is the appropriate approach to correct for multiple testing when the moving averages approach proposed by the same authors is used to identify QTL regions. As such the window average values above the PFP thresholds of 1%, 5% and 10% have been considered as leading QTL average. A 1 log drop in flanking average values defined the boundaries of the QTL region (QTLR). The leading SNP is the one showing the largest −log10 (*p*-value) among those in each QTLR.

### 2.6. Functional Annotation of the QTLR

The SNPchiMp online database [36] was utilized to match the Illumina SNP name with the SNP rsID (Reference SNP cluster ID). The European Variation Archive (EVA) variant browser of EMBL-EBI [37] allowed annotating all the leading SNPs through the rsID. The full genes set (*Bos taurus*: Ensemble Gene 92) was used [38]. Gene ontology (GO) functional annotation and KEGG pathway analyses using the Gene ontology (GO) and pathway analyses were performed using the DAVID Bioinformatics Resources software, version 6.8 [39]. In addition, bovine QTL available from Animal Genome Database [40] were catalogued into our QTLRs by overlapping.

STRING (Search Tool for the Retrieval of Interacting Genes/Proteins) was used to investigate the existence of gene networks in cattle among ones in QTLRs identified with PFP at 10%. Those found in the gene network were annotated by STRING using both bovine and human databases.

## 3. Results

From the 375 animals included in the study, 34% were males, and 65% females, ages 12 to 108 months; 44 months was the most frequent age (22%). From the cases group, lesions were found mainly in lymph nodes of head (retropharyngeal 51%) and thorax (mediastinal and tracheobronchial, 61%), some animals had lesions in more than one lymph node.

### QTLRs Associated with Resistance/Susceptibility to bTB

A total 154 QTLRs at 10% PFP were identified (Figure 2, Appendix A). In general, all these regions were distributed homogeneously over all autosomes (with the exception that none were found on BTA15 and BTA17), and on chromosome X (n. 2), defined by 3296 SNPs. The average length of the QTLRs was 93,446 bp. Appendix A also includes information about the position of the leading SNP for each QTLR on the chromosome, the number of SNPs defining the regions, and its location in the genes annotated within the QTLR, and the number of SNPs pertaining to the regions above each of the three PFP thresholds.

One hundred and seventy-two genes (including 2 miRNA and 5 tRNA) were catalogued in the QTLRs using the *Bos taurus* Ensembl Gene annotation release 92 (Appendix A). The DAVID (The Database for Annotation, Visualization and Integrated Discovery) Database recognized all these genes (excluding miRNA and tRNA), but not for all of them provided the annotated information according to the GO (Gene Ontology) and the KEGG (Kyoto Encyclopedia of Genes and Genomes) pathways terms as in Table 2 (reporting only gene function classifications resulted with a nominal *p* value ≤ 0.05). As shown in Table 2, most genes refer to the immune response and structural terms. Appendix A reports: (i) the list of genes annotated in the QTLRs (list of genes); (ii) the gene annotation according to the DAVID database classification reported as clustered and not clustered genes including those with a nominal *p* value ≥ 0.05.

In Figure 3, the gene network obtained for genes annotated with STRING is shown for *Bos taurus* and *Homo sapiens* proteins. The genes shown are only the ones in QTLRs that were part of a network. Appendix A reports the GO and pathway analysis for the genes included in the networks of Figure 3.

## 4. Discussion

Bovine TB is one of the most prevalent and important diseases in the livestock industry, as well as in wildlife and human population [3]; its eradication is still a priority for many countries. Current strategies to reduce the prevalence in the herds of livestock focus primarily on test-and-disposal of reactors, and abattoir surveillance. In developing countries, however, the success of these programs has been partial because of the poor sensitivity of the tuberculin test and the difficulties tracing back infected animals identified at slaughterhouses. New strategies have been recommended, such as vaccination, in either cattle or the wildlife species [41], estimation of direct genomic estimated breeding values (EBVs) in UK dairy cattle [42], or to increase host resistance through breeding practices [43]. Recent studies in fact have disclosed genetic variability affecting resistance to bTB [20,41,43,44,45,46,47,48] suggesting the possibility of implementing genomic selection for that feature in cattle.

The genes present in the novel QTLRs according to the PFP, 1%, 5% and 10% *p* values threshold were: 

### 4.1. QTLR_1%_PFP

Five QTLRs distributed on different BTA (7, 10, 13, 30 and 31) were identified, but the three genes mapping within these regions are not involved in metabolic pathways associated with bTB.

### 4.2. QTLR_5%_PFP

#### 4.2.1. BTA 1

The QTLR_10 and QTLR_16 include genes involved in immune response to disease. In detail, the first region harbors the TIGIT gene (T cell immunoreceptor with Ig and ITIM (Immunoreceptor tyrosine-based inhibitory motif) domain), an inhibitor of the T cell proliferation, the cytokine production in CD4+ T cells, and of the NK cells cytolytic activity [49]. As reported by Joller et al. [50], the altered balance between activation and inhibitory immune signals can result in increased susceptibility to infection or to induction of autoimmunity.

The *NAALADL2* gene (N-acetylated alpha-linked acidic dipeptidase-like 2), whose function is not well known, is located within the QTLR_16. This gene promotes a pro-migratory and pro-metastatic phenotype in cancer [51], and was recently associated with bovine respiratory disease susceptibility [31].

#### 4.2.2. BTA 3

The *TRIM33* gene (Tripartite motif containing 33), located in QTLR_27 on BTA 3, is involved in migration of macrophages and neutrophils towards inflammatory stimulus in vertebrate tissues [52]. Weng et al. [53] reported *TRIM33* roles in transcriptional regulation during hematopoiesis, tumor suppressor activity in multiple tissues, erythropoiesis, and DNA repair.

#### 4.2.3. BTA 5

On this chromosome, the QTLR_37 harbors the *CD163* gene (CD163 molecule). This gene, expressed on monocytes, macrophages and subpopulations of hematopoietic progenitor is involved in the clearance of haptoglobin–hemoglobin complexes by mediating endocytosis, and prevents the toxic and oxidative effects of free hemoglobin. Different mediators regulate the *CD163* gene expression: up-regulation by glucocorticoids and IL10, and down-regulation by lipopolysaccharide, gamma-interferon, and tumor necrosis factor alpha [54].

#### 4.2.4. BTA 13

Interestingly, from 8.1 Mb to 10.2 Mb, five QTLRs (QTLR_95, QTLR_96, QTLR_97, QTLR_98 and QTLR_99) are found: the first four located within the gene *MACROD2* (MACRO domain containing 2), and the fifth one is in between the end of the gene *MACROD2* and a second gene *KIF16B* (Kinesin Family Member 16 B) (Figure 4). The individual −log10 *p* values show very clear peaks supporting the indication that in this 2 Mb region the *MACROD2* gene and the *KIF16B* gene may play a role in resistance to TB. According to Figure 4 showing the introns and exons of *MACROD2* gene (NCBI refseq gene 105) and the GWAS results for this chromosomal region, the QTLR_98 includes three exons (blue vertical lines). Nevertheless, the *MACROD2* gene is very long and its annotation still needs additional validation [55].

The *KIF16B* gene encodes a kinesin-like protein that could be involved in intracellular trafficking [56].

### 4.3. QTLR_10%_PFP

#### BTA 2

The *DNER* gene (delta/notch-like EGF repeat containing) within the QTLR_21 on BTA 2, is among those differentially expressed for inflammatory diseases, connective tissue disorders and immunological diseases in cattle [57]. In addition, the *DNER* gene expression level has been shown to decrease in highly marbled beef cattle [58].

Six QTLRs harbor seven genes that have been already associated with susceptibility/resistance to TB: QTLR_12 on BTA 1 (*CD80*), QTLR_25 on BTA3 (*CTSS*), QTLR_26 on BTA 3 (*FCGR1A*), QTLR_127 on BTA 23 (*HFE*), QTLR_133 on BTA 25 (*IL21R*), and QTLR_152 on BTA 29 (*ANO9* and *SIGIRR*). These genes are all involved in immune response against *Mycobacterium* spp.

Cathepsins, including Cathepsin S (*CTSS*) are proteolytic enzymes that function mainly in lysosomes, where they contribute to pathogen killing by their involvement in antigen presentation pathways. Pires et al. [59] demonstrated the role of this class of proteins in the control of *M. tuberculosis* by manipulating the cathepsin expression by pathogenic mycobacteria to favor its intracellular survival.The protein encoded by the *CD80* gene (CD80 molecule), the B-lymphocyte activation antigen B7-1 is a membrane receptor that affects the immunological reactivity of T-lymphocytes when its expression decreases. In addition, *CD80* has a role in enhancing the anti-tuberculosis immunity [60].The *FCGR1A* gene (Fc fragment of IgG receptor Ia) expression, together with that of the *BLR1* gene has been considered as potential marker for monitoring the extent of TB disease and to predict treatment outcome in children affected by *M. tuberculosis* [61].Booty et al. [62] reported that the cytokine IL-21, produced predominantly by activated CD4+ T cells and CD8+ T cells, is an essential signaling marker for host resistance to *M. tuberculosis* infection via the IL-21 receptor (IL-21R).Gomes-Pereira et al. [63] reported an increased susceptibility to *M. avium* in Hemochromatosis Protein HFE-Deficient Mice. *HEF* (homeostatic iron regulator) is a fundamental protein involved in the regulation of cellular iron uptake and iron homeostasis. Studies indicate that monocytes with mutated *HFE* have decreased intracellular iron levels [64]. Also, Wang at el. [65] demonstrated that hemochromatosis impacts the regulation of macrophage cytokine translation and, consequently the inflammatory response.The *ANO9* gene, also known as *TMEM16J* (anoctamin 9), together with the *SIGIRR* and the *PKP3* genes constitute a polymorphic complex associated with susceptibility to tuberculosis [66]. The *SIGIRR* gene, also known as Toll IL-1 receptor 8, is a regulatory protein acting to inhibit ILRs and TLRs signaling [67]. The *PKP3*, the third part of this complex gene, maps 6.2 Kb from the end of the QTLR_152.

In addition, three QTLRs identified in our study (QTLR_43 on BTA6, QTLR_115, and QTLR_118 on BTA 21) overlap with those found by Richardson et al. [20] in a study on bovine tuberculosis susceptibility performed in a population of Holstein–Friesian bulls (ID: 96694; 96508, 96511, 96525, 96514, 96517, 96411, 96549, and 96497). Also, the QTLR_84 on BTA12 and QTLR_139 on BTA26 are included within the “Bovine respiratory disease susceptibility QTL (ID: 95663)” and the “Heat tolerance QTL (ID: 31198)”, respectively.

## 5. Conclusions

The results presented here reveal novel QTLRs and confirm mapped loci for resistance to tuberculosis in dairy cattle. The novel QTLRs located on BTA 1, 3, 5, 25 and 29 harbor genes related to immune response. Our results confirm QTL regions previously mapped on BTA 2, 6, 13, 21, 22 and 23 related to resistance to bTB in other dairy cattle populations.

Genomic regions and genes identified in the present study with a case-control selective DNA pooling approach were significantly associated with resistance to TB in cattle. The findings of this study could be used to improve the knowledge on the bTB immune response against *Mycobacterium bovis*, and thus provide the basis for genetic control of this disease in cattle.

## Figures and Tables

**Figure 1 animals-09-00636-f001:**
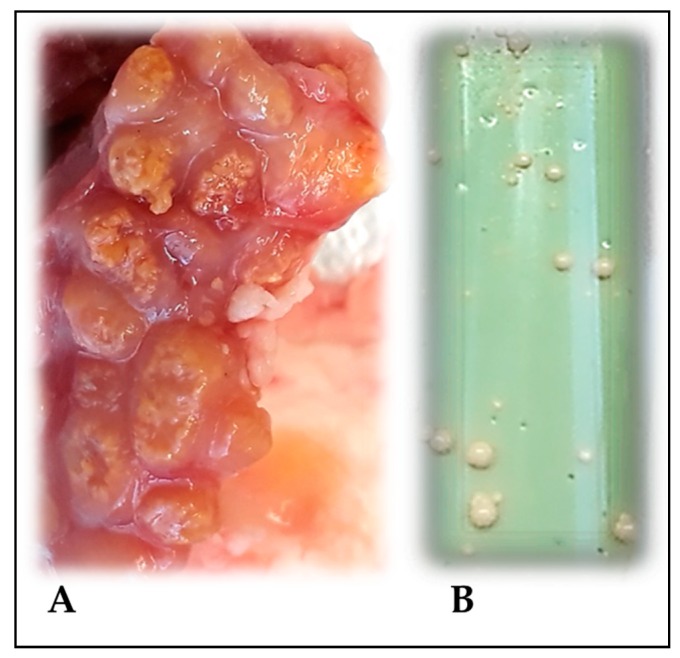
Tracheobronchial lymph nodes with visible lesions of Bovine tuberculosis (bTB) (**A**). Colonies of *M. bovis* in Stonebrink media (**B**).

**Figure 2 animals-09-00636-f002:**
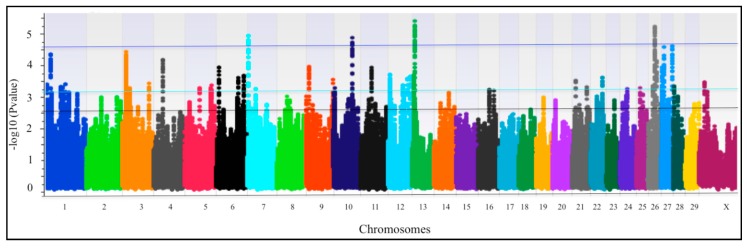
Manhattan plots of QTLR (Quantitative Trait Loci Regions) for all chromosomes. Horizontal lines represent the 1% PFP (proportion of false positives) (blue), the 5% PFP (light blue), and the 10% PFP (black) thresholds.

**Figure 3 animals-09-00636-f003:**
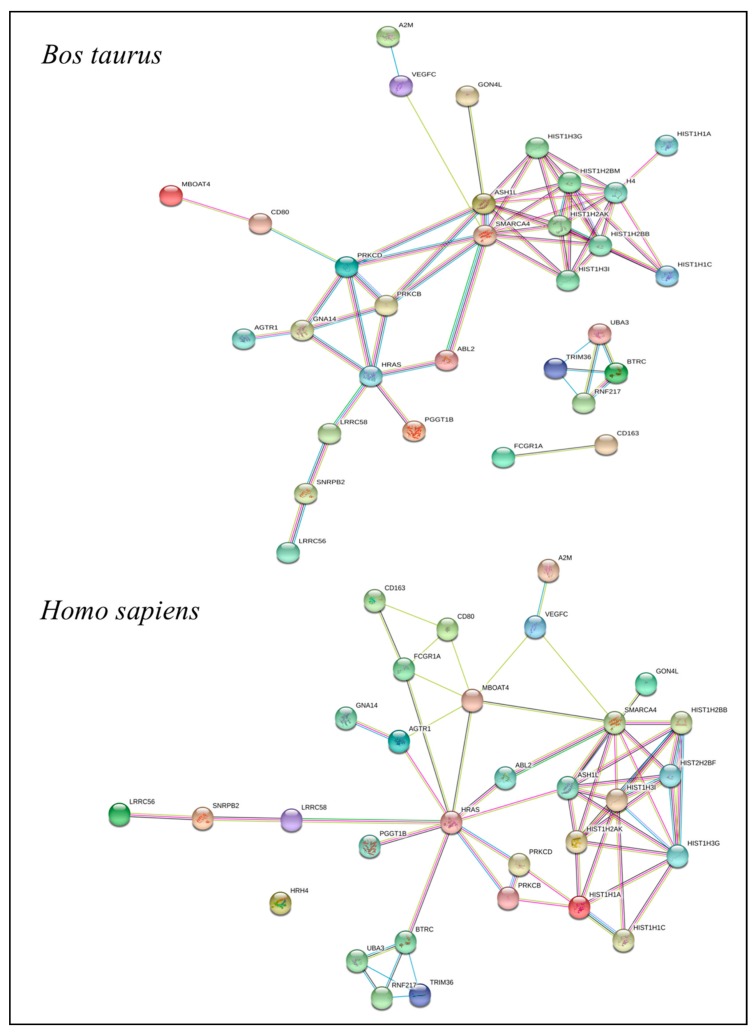
Gene networks in cattle among the ones in QTLRs identified with PFP at 10% (Bovine and Human databases).

**Figure 4 animals-09-00636-f004:**
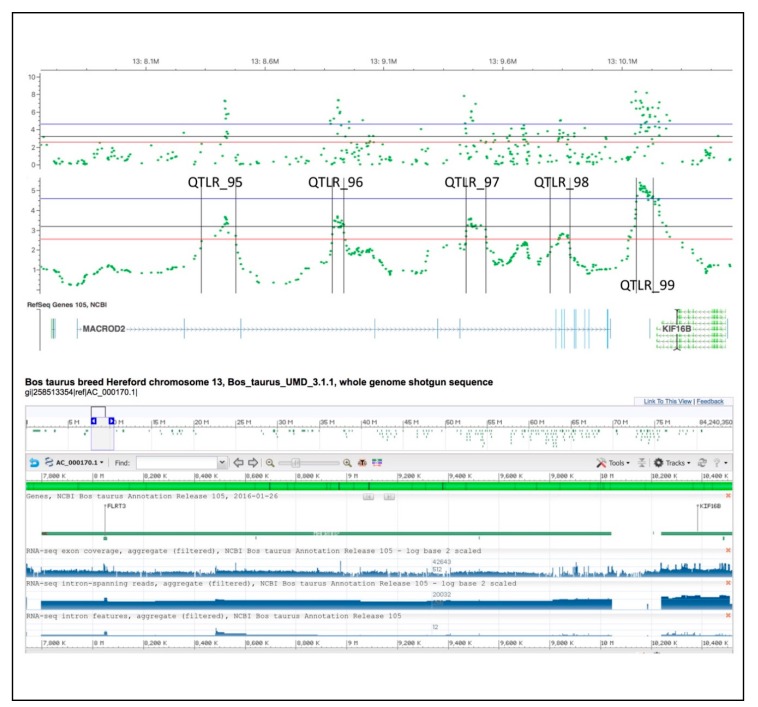
The location of MACROD2 gene is spread over the QTLRs 95–98 (NCBI refseq gene 105).

**Table 1 animals-09-00636-t001:** Scheme of case (CA) and controls (CT) pool definitions, and of genotyping.

CASES			CONTROLS		
Biological	Technical		Biological	Technical	
Bio_rep ^a^	Pool_Rep ^b^	Array_Rep ^c^	Bio_rep ^a^	Pool_Rep ^b^	Array_Rep ^c^
CA_1	CA1_A	CA_1A_1	CT_1	CT1_A	CT_1A_1
		CA_1A_2			CT_1A_2
		CA_1A_3			CT_1A_3
	CA1_B	CA_1B_1		CT1_B	CT_1B_1
		CA_1B_2			CT_1B_2
		CA_1B_3			CT_1B_3
CA_2	CA2_A	CA_2A_1	CT_2	CT2_A	CT_2A_1
		CA_2A_2			CT_2A_2
		CA_2A_3			CT_2A_3
	CA2_B	CA_2B_1		CT2_B	CT_2B_1
		CA_2B_2			CT_2B_2
		CA_2B_3			CT_2B_3
			CT_3	CT3_A	CT_3A_1
					CT_3A_3
					CT_3A_3
				CT3_B	CT_3B_1
					CT_3B_3
					CT_3B_3

^a^ Bio_rep = biological replicate; ^b^ Pool_Rep = technical pooling replicate; ^c^ Array_Rep = technical array replicate.

**Table 2 animals-09-00636-t002:** Results of the gene annotation: DAVID GO and pathway analysis (KEGG).

Term	Count	*p*-Value	Genes
Biological process			
GO:0006334: nucleosome assembly	6	9.42 × 10^5^	*HIST1H2BB*, *HIST1H1C*, *HIST1H1A*, *H2B*, *HIST1H3G*, HIST1H3I
GO:0006335: DNA replication-dependent nucleosome assembly	3	6.12 × 10^3^	*H4*, *HIST1H3G*, *HIST1H3I*
GO:0051290: protein heterotetramerization	3	6.81 × 10^3^	*H4*, *HIST1H3G*, *HIST1H3I*
GO:0098792: xenophagy	5	2.27 × 10^3^	*TMEM39A*, *SNRPB2*, *CPA3*, *HIST1H3G*, *HIST1H3I*
GO:0002230: positive regulation of defense response to virus by host	5	3.68 × 10^3^	*TMEM39A*, *SNRPB2*, *CPA3*, *HIST1H3G*, *HIST1H3I*
GO:0046627: negative regulation of insulin receptor signaling pathway	3	9.09 × 10^3^	*PRKCD*, *KANK1*, *PRKCB*
GO:0042742: defense response to bacterium	4	1.63 × 10^2^	*STAB1*, *FCGR1A*, *PRKCD*, *TMF1*
Cellular Components			
GO:0000786: nucleosome	7	1.60 × 10^5^	*H4*, *HIST1H1C*, *HIST1H1A*, *H2B*, *HIST1H2AK*, *HIST1H3G*, *HIST1H3I*
GO:0000788: nuclear nucleosome	5	2.47 × 10^4^	*HIST1H2BB*, *H2B*, *HIST1H3G*, *HIST1H3I*
GO:0000784: nuclear chromosome, telomeric region	4	2.3 × 10^2^	*H4*, *TNKS*, *HIST1H3G*, *HIST1H3I*
GO:0030176: integral component of endoplasmic reticulum membrane	4	1.50 × 10^2^	*PIGG*, *SARAF*, *MBOAT4*, *SLC27A2*
GO:0005615: extracellular space	13	3.32 × 10^2^	*A2M*, *H2B*, *HFE*, *FSTL1*, *CTSS*, *OVOS2*, *ESF1*, *VEGFC*, *GPI*, *CTSK*, *CPA3*, *CPB1*, *SMARCA4*
GO:0005788: endoplasmic reticulum lumen	3	4.77 × 10^2^	*EOGT*, *SLC27A2*, *POGLUT1*
Molecular Functions			
GO:0046982: protein heterodimerization activity	5	7.15 × 10^3^	*AGTR1*, *HIST1H2BB*, *H4*, *H2B*, *FOXP1*
GO:0042393: histone binding	3	3.67 × 10^2^	*H4*, *PRKCB*, *SMARCA4*
KEGG Pathways			
bta05322: Systemic lupus erythematosus	9	7.64 × 10^8^	*HIST1H2BB*, *H4*, *CD80*, *FCGR1A*, *HIST2H2BF*, *H2B*, *HIST1H2AK*, *HIST1H3G*, *HIST1H3I*
bta05034: Alcoholism	8	4.32 × 10^6^	*HIST1H2BB*, *HRAS*, *H4*, *HIST2H2BF*, *H2B*, *HIST1H2AK*, *HIST1H3G*, *HIST1H3I*
bta05203: Viral carcinogenesis	5	1.92 × 10^3^	*HIST1H2BB*, *HRAS*, *H4*, *HIST2H2BF*, *H2B*
bta00514: Other types of O-glycan biosynthesis	3	2.18 × 10^2^	*ST6GAL2*, *EOGT*, *POGLUT1*

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
