# Peer review of "Genome-Wide Association Study in Mexican Holstein Cattle Reveals Novel Quantitative Trait Loci Regions and Confirms Mapped Loci for Resistance to Bovine Tuberculosis"

_animals, 2019, doi:10.3390/ani9090636_

Round 1

Reviewer 1 Report

The manuscript describes a successful work on the identification of the bovine tuberculosis resistance determinants in the local population of Holstein cattle.

In spite of comparatively frequent papers reporting genome-wide association studies on the resistance to bovine tuberculosis or paratuberculosis, three factors probably contributed to the success of the reported study. First, the case-control arrangement of the study enabled efficient use of financial means and laboratory capacity. Second, the use of a high density chip instead of the standard 54K bovine arrays increased the statistical power of the study. Third, the endemic spread of bovine tuberculosis in the included populations presumably provided uniform exposure of all animals, both susceptible and resistant, to the infection. Unequal exposure to the disease, usually a consequence of intense surveillance and low incidence, is a limiting factor for genome-wide association studies (Finlay et al., PLoS ONE 7, e30545, 2012). Moreover, direct scoring of the post mortem diagnoses allows to avoid inherent incorrectness of the otherwise prevailing tuberculin test in phenotyping.

The results provided a comprehensive picture of resistance factors against tuberculosis present in the Holstein cattle gene pool. It is indicative that many genes identified in the study matched the genes or QTLs previously reported in relation to the tuberculosis resistance. This provides a feedback on the reliability of the obtained results. Although the multiple comparison approach is inevitably associated with a certain proportion of false discoveries (FDR), its role in the present study seems to remain under the acceptable limit.

The most points to be corrected in the manuscript are the formulations that sometimes suffer from the oversights in English. I attach a PDF file of the manuscript with indicated errors in the language or with formulations that might be improved. The use of these comments depends on the consideration of the authors.

Also the number of genes analysed with KEGG in Table 2 (n = 108) differs form the number indicated for Table S2 (n = 172) in the beginning of the paragraph on l. 213.

The denotation of genes in Fig. 3 will not withstand reproduction.

I suggest to include the sentence with this approximate content ”The biology of the identified genes provides additional supporting evidence for the conclusions based solely on the GWAS statistics” in the beginning of the Discussion, presumably on line 241. The sentence would play a role of introduction of the systemic part of the Discussion interpreting the candidate genes in the chromosome order.

I have a pleasure to recommend the manuscript for publication.

Author Response

Manuscript title: Genome-wide association study in Mexican Holstein reveals novel QTL regions and confirms mapped loci for resistance to bovine tuberculosis.

------------------------------------------------------------------------------------------------------------------------------------

Reviewer 1

Comments and Suggestions for Authors

The manuscript describes a successful work on the identification of the bovine tuberculosis resistance determinants in the local population of Holstein cattle.

In spite of comparatively frequent papers reporting genome-wide association studies on the resistance to bovine tuberculosis or paratuberculosis, three factors probably contributed to the success of the reported study. First, the case-control arrangement of the study enabled efficient use of financial means and laboratory capacity. Second, the use of a high density chip instead of the standard 54K bovine arrays increased the statistical power of the study. Third, the endemic spread of bovine tuberculosis in the included populations presumably provided uniform exposure of all animals, both susceptible and resistant, to the infection. Unequal exposure to the disease, usually a consequence of intense surveillance and low incidence, is a limiting factor for genome-wide association studies (Finlay et al., PLoS ONE 7, e30545, 2012). Moreover, direct scoring of the post mortem diagnoses allows to avoid inherent incorrectness of the otherwise prevailing tuberculin test in phenotyping.

The results provided a comprehensive picture of resistance factors against tuberculosis present in the Holstein cattle gene pool. It is indicative that many genes identified in the study matched the genes or QTLs previously reported in relation to the tuberculosis resistance. This provides a feedback on the reliability of the obtained results. Although the multiple comparison approach is inevitably associated with a certain proportion of false discoveries (FDR), its role in the present study seems to remain under the acceptable limit.

Reviewer comment 1:

The most points to be corrected in the manuscript are the formulations that sometimes suffer from the oversights in English. I attach a PDF file of the manuscript with indicated errors in the language or with formulations that might be improved. The use of these comments depends on the consideration of the authors.

Authors reply:  All comments from the reviewer were considered important and therefore included in the manuscript.

Reviewer comment 2:

Also the number of genes analysed with KEGG in Table 2 (n = 108) differs form the number indicated for Table S2 (n = 172) in the beginning of the paragraph on l. 213.

Authors reply:  The correction was made, the parragraph ended up as follows.

One hundred and seventy-two genes (including 2 miRNA and 5 tRNA) were catalogued in the QTLRs using the gene Bos taurus Ensembl Gene annotation release 92 (Table S2). DAVID Database recognized all these genes (excluding miRNA and tRNA), but not for all of them provided the annotated information according to the GO and the KEGG pathways terms as in Table 2 (reporting only gene function classifications resulted with a nominal P value ≤ 0.05).

Reviewer comment 3:

The denotation of genes in Fig. 3 will not withstand reproduction.

Authors reply:  The format of Fig. 3 was corrected.

I suggest to include the sentence with this approximate content ”The biology of the identified genes provides additional supporting evidence for the conclusions based solely on the GWAS statistics” in the beginning of the Discussion, presumably on line 241. The sentence would play a role of introduction of the systemic part of the Discussion interpreting the candidate genes in the chromosome order.

Authors reply:  Done as suggested.

I have a pleasure to recommend the manuscript for publication.

Reviewer 2 Report

Manuscript title: Genome-wide association study in Mexican Holstein reveals novel QTL regions and confirms mapped loci for resistance to bovine tuberculosis

The authors applied a selective genotyping method, generally used to reduce costs, to a Mexican dairy cattle population to investigate the possible genetic host susceptibility/resistance to Bovine tuberculosis.

Please find some comments/suggestions below:

Line 64-67: The disease…in the herds. Please check the grammar of this statement, there is something wrong. What I understand is that even if there are individual checks at the abattoirs and random herd checks at the time of the slaughter, the disease is still persistent

Line 78-82 There is a paper using Italian Holstein. Please cite and include Minozzi G, Buggiotti L, Stella A, Strozzi F, Luini M, Williams JL. Genetic loci involved in antibody response to Mycobacterium avium ssp. paratuberculosis in cattle. PLoS One. 2010;5(6):e11117. Published 2010 Jun 15. doi:10.1371/journal.pone.0011117

Line 78-82. In UK a Genetic Evaluation for bTB is now available. (G. Banos, M. Winters, R. Mrode, A.P. Mitchell, S.C. Bishop, J.A. Woolliams, M.P. Coffey, Genetic evaluation for bovine tuberculosis resistance in dairy cattle, Journal of Dairy Science, Volume 100, Issue 2, 2017, Pages 1272-1281, ISSN 0022-0302, https://doi.org/10.3168/jds.2016-11897. ) I think it’s worthwhile to report it here or in the discussion session to suggest that selection for “resistance” is possible (discussion session  - line 231-232)

MM section

Regarding the experimental design, even if I do understand that the samples are actually coming from commercial abattoirs, I have some concerns about possible hidden stratification which can have biased your results. 

For example both breed composition and age have been previously identified as important effect in bTB but there are no detailed information about the characteristics of the two experimental groups. 

We only know, for instance, that 33% were males and 63% females but we do not know the actual proportion in case and controls. 

We do not even know the average age in cases and controls. 

Furthermore, breed composition is another issue. Probably it is nearly impossible to know the effective breed composition but what about if the two experimental groups are unbalanced? Can you speculate about it? Additionally, if non purebred Holsteins were included your title is not correct. Indeed,  you should more generally say Mexican Dairy Cattle

Would It be possible to add a table with some more accurate statistics on cases and controls?

Line 92:  Animals slaughtered were mainly Holstein... What does "mainly" means? see my comments about breed composition. 

Line 175 Please define PFP (Proportion of Falses Positives?)

Line 203 (A total 154 QTLRs at 10% PFP were identified) and Line 205-206 (From the total QTLR found, 154 were at the 10% PFP, 42 at 5% PFP and 5 at 1% PFP) are partially saying the same thing. Please rephrase

Discussion Section

A general important comment regards the use of the results of a GWAS: a GWAS suggest the difference in frequency of some alleles in cases and controls (obtained using more or less complicated statistical approaches). Actually it does not say the direction. Some alleles are for resistance and others for susceptibility. In both the Discussion and Conclusions sections you always talk about resistance and I think that this is not completely correct.

For instance, you say that one of the novel QTLRs is located on BTA1 including genes involved in immune response to disease (line 247). This sounds interesting, but immune response can affect susceptibility or resistance.  Indeed, Jolley reported which can increase susceptibility (not resistance, actually). The NAALADL2 gene, is associated with resistance or susceptibility ? So, the question is : resistance or susceptibility? Probably both and for this reason you should be careful in your conclusions and in your title as well.

Line 231 – 232 Genetic selection tools like the UK EBVs are examples of additional strategies. It would be worthwhile to add it to the discussion section

Line 387: severuty=severity

Author Response

Manuscript title: Genome-wide association study in Mexican Holstein reveals novel QTL regions and confirms mapped loci for resistance to bovine tuberculosis.

------------------------------------------------------------------------------------------------------------------------------------

Reviewer 2

Comments and Suggestions for Authors

The authors applied a selective genotyping method, generally used to reduce costs, to a Mexican dairy cattle population to investigate the possible genetic host susceptibility/resistance to Bovine tuberculosis.

Please find some comments/suggestions below:

Reviewer comment 1:

Line 64-67: The disease…in the herds. Please check the grammar of this statement, there is something wrong. What I understand is that even if there are individual checks at the abattoirs and random herd checks at the time of the slaughter, the disease is still persistent

Authors reply:  The parragraph was rewrited to clarify the idea.

Reviewer comment 2:

Line 78-82 There is a paper using Italian Holstein. Please cite and include Minozzi G, Buggiotti L, Stella A, Strozzi F, Luini M, Williams JL. Genetic loci involved in antibody response to Mycobacterium avium ssp. paratuberculosis in cattle. PLoS One. 2010;5(6):e11117. Published 2010 Jun 15. doi:10.1371/journal.pone.0011117

Authors reply: Information from this paper was included in the manuscript´s text.

Reviewer comment 3:

Line 78-82. In UK a Genetic Evaluation for bTB is now available. (G. Banos, M. Winters, R. Mrode, A.P. Mitchell, S.C. Bishop, J.A. Woolliams, M.P. Coffey, Genetic evaluation for bovine tuberculosis resistance in dairy cattle, Journal of Dairy Science, Volume 100, Issue 2, 2017, Pages 1272-1281, ISSN 0022-0302, https://doi.org/10.3168/jds.2016-11897. ) I think it’s worthwhile to report it here or in the discussion session to suggest that selection for “resistance” is possible (discussion session - line 231-232)

Authors reply:  The suggested paper was included in the manuscript.

MM section

Reviewer comment 4:

Regarding the experimental design, even if I do understand that the samples are actually coming from commercial abattoirs, I have some concerns about possible hidden stratification which can have biased your results. 

For example both breed composition and age have been previously identified as important effect in bTB but there are no detailed information about the characteristics of the two experimental groups. 

We only know, for instance, that 33% were males and 63% females but we do not know the actual proportion in case and controls. 

We do not even know the average age in cases and controls. 

Authors reply:  We agree with the reviewer concerns in relation to information about the animals missing; whoever, we know that all dairy cattle in the sampled region have a high level of genetic similarity since all replacements are produced in the same farms, all of them are 100% Holstein. The average age and the proportion of males and females per cases and controls were included.

Reviewer comment 5:

Furthermore, breed composition is another issue. Probably it is nearly impossible to know the effective breed composition but what about if the two experimental groups are unbalanced? Can you speculate about it? Additionally, if non purebred Holsteins were included your title is not correct. Indeed, you should more generally say Mexican Dairy Cattle

Authors reply: We believe that the actual title is the best according to the material in the manuscript.

Reviewer comment 6:

Would it be possible to add a table with some more accurate statistics on cases and controls?

Authors reply: The average age and standard error for cases and controls were added in the manuscript.  

Reviewer comment 7:

Line 92:  Animals slaughtered were mainly Holstein... What does "mainly" means? see my comments about breed composition. 

Authors reply:  We decided to delete the Word “mainly” to prevent any confusion.

Reviewer comment 8:

Line 175 Please define PFP (Proportion of Falses Positives?)

Authors reply: Done 

Reviewer comment 9:

Line 203 (A total 154 QTLRs at 10% PFP were identified) and Line 205-206 (From the total QTLR found, 154 were at the 10% PFP, 42 at 5% PFP and 5 at 1% PFP) are partially saying the same thing. Please rephrase

Authors reply:  Done

Discussion Section

A general important comment regards the use of the results of a GWAS: a GWAS suggest the difference in frequency of some alleles in cases and controls (obtained using more or less complicated statistical approaches). Actually it does not say the direction. Some alleles are for resistance and others for susceptibility. In both the Discussion and Conclusions sections you always talk about resistance and I think that this is not completely correct.

Reviewer comment 10:

For instance, you say that one of the novel QTLRs is located on BTA1 including genes involved in immune response to disease (line 247). This sounds interesting, but immune response can affect susceptibility or resistance.  Indeed, Jolley reported which can increase susceptibility (not resistance, actually). The NAALADL2 gene, is associated with resistance or susceptibility ? So, the question is : resistance or susceptibility? Probably both and for this reason you should be careful in your conclusions and in your title as well.

Authors reply:  We agree with the reviewer comment, this point is somewhat controversial. The word “susceptibility” was included in the text since the genes involved are associated to immune response; therefore, they can be related to resistance or to susceptibility.

Reviewer comment 11:

Line 231 – 232 Genetic selection tools like the UK EBVs are examples of additional strategies. It would be worthwhile to add it to the discussion section

Authors reply:  In the discussion section we included the tools for the genetic selection based on EBVs.

Reviewer comment 11:

Line 387: severuty=severity

Authors reply: Done  

Round 2

Reviewer 2 Report

I would like to thank the authors for accepting suggestions and clarifying some aspects. 

A very last issue to be clarified regards the proportion of males and females in both cases and controls which have been added at lines 139-140 and 142-143. Are they correct? I guess that there is something wrong with males proportion: it should be 24 and 38 % for cases and controls, respectively.

Author Response

Manuscript title: Genome-wide association study in Mexican Holstein reveals novel QTL regions and confirms mapped loci for resistance to bovine tuberculosis.

------------------------------------------------------------------------------------------------------------------------------------

Reviewer 1

Comments and Suggestions for Authors

A very last issue to be clarified regards the proportion of males and females in both cases and controls which have been added at lines 139-140 and 142-143. Are they correct? I guess that there is something wrong with males proportion: it should be 24 and 38 % for cases and controls, respectively.

The Comments and Suggestions have been done.